# Barzilai-Borwein Step Size for Stochastic Gradient Descent

**Conghui Tan**
The Chinese University of Hong Kong
chtan@se.cuhk.edu.hk

**Shiqian Ma**
The Chinese University of Hong Kong
sqma@se.cuhk.edu.hk

**Yu-Hong Dai**
Chinese Academy of Sciences, Beijing, China
dyh@lsec.cc.ac.cn

**Yuqiu Qian**
The University of Hong Kong
qyq79@connect.hku.hk

## Abstract

One of the major issues in stochastic gradient descent (SGD) methods is how to choose an appropriate step size while running the algorithm. Since the traditional line search technique does not apply for stochastic optimization methods, the common practice in SGD is either to use a diminishing step size, or to tune a step size by hand, which can be time consuming in practice. In this paper, we propose to use the Barzilai-Borwein (BB) method to automatically compute step sizes for SGD and its variant: stochastic variance reduced gradient (SVRG) method, which leads to two algorithms: SGD-BB and SVRG-BB. We prove that SVRG-BB converges linearly for strongly convex objective functions. As a by-product, we prove the linear convergence result of SVRG with *Option I* proposed in [10], whose convergence result has been missing in the literature. Numerical experiments on standard data sets show that the performance of SGD-BB and SVRG-BB is comparable to and sometimes even better than SGD and SVRG with best-tuned step sizes, and is superior to some advanced SGD variants.

## 1 Introduction

The following optimization problem, which minimizes the sum of cost functions over samples from a finite training set, appears frequently in machine learning:

$$\min \ F(x) \equiv \frac{1}{n} \sum_{i=1}^{n} f_i(x), \tag{1}$$

where $n$ is the sample size, and each $f_i : \mathbb{R}^d \to \mathbb{R}$ is the cost function corresponding to the $i$-th sample data. Throughout this paper, we assume that each $f_i$ is convex and differentiable, and the function $F$ is strongly convex. Problem (1) is challenging when $n$ is extremely large so that computing $F(x)$ and $\nabla F(x)$ for given $x$ is prohibited. Stochastic gradient descent (SGD) method and its variants have been the main approaches for solving (1). In the $t$-th iteration of SGD, a random training sample $i_t$ is chosen from $\{1, 2, \ldots, n\}$ and the iterate $x_t$ is updated by

$$x_{t+1} = x_t - \eta_t \nabla f_{i_t}(x_t), \tag{2}$$

where $\nabla f_{i_t}(x_t)$ denotes the gradient of the $i_t$-th component function at $x_t$, and $\eta_t > 0$ is the step size (a.k.a. learning rate). In (2), it is usually assumed that $\nabla f_{i_t}$ is an unbiased estimation to $\nabla F$, i.e.,

$$\mathbb{E}[\nabla f_{i_t}(x_t) \mid x_t] = \nabla F(x_t). \tag{3}$$

However, it is known that the total number of gradient evaluations of SGD depends on the variance of the stochastic gradients and it is of sublinear convergence rate for strongly convex and smooth problem (1). As a result, many works along this line have been focusing on designing variants of SGD that can reduce the variance and improve the complexity. Some popular methods include the stochastic average gradient (SAG) method [16], the SAGA method [7], the stochastic dual coordinate ascent (SDCA) method [17], and the stochastic variance reduced gradient (SVRG) method [10]. These methods are proven to converge linearly on strongly convex problems.

As pointed out by Le Roux et al. [16], one important issue regarding to stochastic algorithms that has not been fully addressed in the literature, is how to choose an appropriate step size $\eta_t$ while running the algorithm. In classical gradient descent method, the step size is usually obtained by employing line search techniques. However, line search is computationally prohibited in stochastic gradient methods because one only has sub-sampled information of function value and gradient. As a result, for SGD and its variants used in practice, people usually use a diminishing step size $\eta_t$, or use a best-tuned fixed step size. Neither of these two approaches can be efficient.

Some recent works that discuss the choice of step size in SGD are summarized as follows. AdaGrad [8] scales the gradient by the square root of the accumulated magnitudes of the gradients in the past iterations, but this still requires to decide a fixed step size $\eta$. [16] suggests a line search technique on the component function $f_{i_k}(x)$ selected in each iteration, to estimate step size for SAG. [12] suggests performing line search for an estimated function, which is evaluated by a Gaussian process with samples $f_{i_t}(x_t)$. [13] suggests to generate the step sizes by a given function with an unknown parameter, and to use the online SGD to update this unknown parameter.

**Our contributions** in this paper are in several folds.
(i) We propose to use the Barzilai-Borwein (BB) method to compute the step size for SGD and SVRG. The two new methods are named as SGD-BB and SVRG-BB, respectively. The per-iteration computational cost of SGD-BB and SVRG-BB is almost the same as SGD and SVRG, respectively.
(ii) We prove the linear convergence of SVRG-BB for strongly convex function. As a by-product, we show the linear convergence of SVRG with Option I (SVRG-I) proposed in [10]. Note that in [10] only convergence of SVRG with Option II (SVRG-II) was given, and the proof for SVRG-I has been missing in the literature. However, SVRG-I is numerically a better choice than SVRG-II, as demonstrated in [10].
(iii) We conduct numerical experiments for SGD-BB and SVRG-BB on solving logistic regression and SVM problems. The numerical results show that SGD-BB and SVRG-BB are comparable to and sometimes even better than SGD and SVRG with best-tuned step sizes. We also compare SGD-BB with some advanced SGD variants, and demonstrate that our method is superior.

The rest of this paper is organized as follows. In Section 2 we briefly introduce the BB method in the deterministic setting. In Section 3 we propose our SVRG-BB method, and prove its linear convergence for strongly convex function. As a by-product, we also prove the linear convergence of SVRG-I. In Section 4 we propose our SGD-BB method. A smoothing technique is also implemented to improve the performance of SGD-BB. Finally, we conduct some numerical experiments for SVRG-BB and SGD-BB in Section 5.

## 2  The Barzilai-Borwein Step Size

The BB method, proposed by Barzilai and Borwein in [2], has been proven to be very successful in solving nonlinear optimization problems. The key idea behind the BB method is motivated by quasi-Newton methods. Suppose we want to solve the unconstrained minimization problem

$$\min_x \; f(x), \tag{4}$$

where $f$ is differentiable. A typical iteration of quasi-Newton methods for solving (4) is:

$$x_{t+1} = x_t - B_t^{-1} \nabla f(x_t), \tag{5}$$

where $B_t$ is an approximation of the Hessian matrix of $f$ at the current iterate $x_t$. The most important feature of $B_t$ is that it must satisfy the so-called secant equation: $B_t s_t = y_t$, where $s_t = x_t - x_{t-1}$ and $y_t = \nabla f(x_t) - \nabla f(x_{t-1})$ for $t \geq 1$. It is noted that in (5) one needs to solve a linear system, which may be time consuming when $B_t$ is large and dense.

One way to alleviate this burden is to use the BB method, which replaces $B_t$ by a scalar matrix $(1/\eta_t)I$. However, one cannot choose a scalar $\eta_t$ such that the secant equation holds with $B_t = (1/\eta_t)I$. Instead, one can find $\eta_t$ such that the residual of the secant equation, i.e., $\|(1/\eta_t)s_t - y_t\|_2^2$, is minimized, which leads to the following choice of $\eta_t$:

$$\eta_t = \|s_t\|_2^2 / \left( s_t^\top y_t \right). \tag{6}$$

Therefore, a typical iteration of the BB method for solving (4) is

$$x_{t+1} = x_t - \eta_t \nabla f(x_t), \tag{7}$$

where $\eta_t$ is computed by (6).

For convergence analysis, generalizations and variants of the BB method, we refer the interested readers to [14, 15, 6, 9, 4, 5, 3] and references therein. Recently, BB method has been successfully applied for solving problems arising from emerging applications, such as compressed sensing [21], sparse reconstruction [20] and image processing [19].

## 3   Barzilai-Borwein Step Size for SVRG

We see from (7) and (6) that the BB method does not need any parameter and the step size is computed while running the algorithm. This has been the main motivation for us to work out a black-box stochastic gradient descent method that can compute the step size automatically without requiring any parameters. In this section, we propose to incorporate the BB step size to SVRG, which leads to the SVRG-BB method.

### 3.1   SVRG-BB Method

Stochastic variance reduced gradient (SVRG) is a variant of SGD proposed in [10], which utilizes a variance reduction technique to alleviate the impact of the random samplings of the gradients. SVRG computes the full gradient $\nabla F(x)$ of (1) in every $m$ iterations, where $m$ is a pre-given integer, and the full gradient is then used for generating stochastic gradients with lower variance in the next $m$ iterations (the next epoch). In SVRG, the step size $\eta$ needs to be provided by the user. According to [10], the choice of $\eta$ depends on the Lipschitz constant of $F$, which is usually difficult to estimate in practice.

Our SVRG-BB algorithm is described in Algorithm 1. The only difference between SVRG and SVRG-BB is that in the latter we use BB method to compute the step size $\eta_k$, instead of using a prefixed $\eta$ as in SVRG.

---

**Algorithm 1** SVRG with BB step size (SVRG-BB)

---

**Parameters**: update frequency $m$, initial point $\tilde{x}_0$, initial step size $\eta_0$ (only used in the first epoch)
**for** $k = 0, 1, \cdots$ **do**
$\quad g_k = \frac{1}{n} \sum_{i=1}^{n} \nabla f_i(\tilde{x}_k)$
$\quad$**if** $k > 0$ **then**
$\quad\quad \eta_k = \frac{1}{m} \cdot \|\tilde{x}_k - \tilde{x}_{k-1}\|_2^2 / (\tilde{x}_k - \tilde{x}_{k-1})^\top (g_k - g_{k-1})$ $\qquad\qquad(\triangle)$
$\quad$**end if**
$\quad x_0 = \tilde{x}_k$
$\quad$**for** $t = 0, \cdots, m-1$ **do**
$\quad\quad$Randomly pick $i_t \in \{1, \ldots, n\}$
$\quad\quad x_{t+1} = x_t - \eta_k(\nabla f_{i_t}(x_t) - \nabla f_{i_t}(\tilde{x}_k) + g_k)$
$\quad$**end for**
$\quad$**Option I**: $\tilde{x}_{k+1} = x_m$
$\quad$**Option II**: $\tilde{x}_{k+1} = x_t$ for randomly chosen $t \in \{1, \ldots, m\}$
**end for**

---

**Remark 1.** *A few remarks are in demand for the SVRG-BB algorithm.*
*(i) If we always set $\eta_k = \eta$ in SVRG-BB instead of using $(\triangle)$, then it reduces to the original SVRG.*
*(ii) One may notice that $\eta_k$ is equal to the step size computed by the BB formula (6) divided by $m$. This is because in the inner loop for updating $x_t$, $m$ unbiased gradient estimators are added to $x_0$ to*

get $x_m$.

*(iii) For the first epoch of SVRG-BB, a step size $\eta_0$ needs to be specified. However, we observed from our numerical experiments that the performance of SVRG-BB is not sensitive to the choice of $\eta_0$.*
*(iv) The BB step size can also be naturally incorporated to other SVRG variants, such as SVRG with batching [1].*

## 3.2  Linear Convergence Analysis

In this section, we analyze the linear convergence of SVRG-BB (Algorithm 1) for solving (1) with strongly convex objective $F(x)$, and as a by-product, our analysis also proves the linear convergence of SVRG-I. The proofs in this section are provided in the supplementary materials. The following assumption is made throughout this section.

**Assumption 1.** *We assume that (3) holds for any $x_t$. We assume that the objective function $F(x)$ is $\mu$-strongly convex, i.e.,*

$$F(y) \geq F(x) + \nabla F(x)^\top (y - x) + \frac{\mu}{2}\|x - y\|_2^2, \quad \forall x, y \in \mathbb{R}^d.$$

*We also assume that the gradient of each component function $f_i(x)$ is $L$-Lipschitz continuous, i.e.,*

$$\|\nabla f_i(x) - \nabla f_i(y)\|_2 \leq L\|x - y\|_2, \ \forall x, y \in \mathbb{R}^d.$$

*Under this assumption, it is easy to see that $\nabla F(x)$ is also $L$-Lipschitz continuous.*

We first provide the following lemma, which reveals the relationship between the distances of two consecutive iterates to the optimal point.

**Lemma 1.** *Define*

$$\alpha_k := (1 - 2\eta_k\mu(1 - \eta_k L))^m + \frac{4\eta_k L^2}{\mu(1 - \eta_k L)}. \tag{8}$$

*For both SVRG-I and SVRG-BB, we have the following inequality for the $k$-th epoch:*

$$\mathbb{E}\|\tilde{x}_{k+1} - x^*\|_2^2 < \alpha_k\|\tilde{x}_k - x^*\|_2^2,$$

*where $x^*$ is the optimal solution to (1).*

The linear convergence of SVRG-I follows immediately.

**Corollary 1.** *In SVRG-I, if $m$ and $\eta$ are chosen such that*

$$\alpha := (1 - 2\eta\mu(1 - \eta L))^m + \frac{4\eta L^2}{\mu(1 - \eta L)} < 1, \tag{9}$$

*then SVRG-I converges linearly in expectation:*

$$\mathbb{E}\|\tilde{x}_k - x^*\|_2^2 < \alpha^k\|\tilde{x}_0 - x^*\|_2^2.$$

**Remark 2.** *We now give some remarks on this convergence result.*
*(i) To the best of our knowledge, this is the first time that the linear convergence of SVRG-I is established.*
*(ii) The convergence result given in Corollary 1 is for the iterates $\tilde{x}_k$, while the one given in [10] is for the objective function values $F(\tilde{x}_k)$.*

The following theorem establishes the linear convergence of SVRG-BB (Algorithm 1).

**Theorem 1.** *Denote $\theta = (1 - e^{-2\mu/L})/2$. Note that $\theta \in (0, 1/2)$. In SVRG-BB, if $m$ is chosen such that*

$$m > \max\left\{\frac{2}{\log(1 - 2\theta) + 2\mu/L}, \frac{4L^2}{\theta\mu^2} + \frac{L}{\mu}\right\}, \tag{10}$$

*then SVRG-BB (Algorithm 1) converges linearly in expectation:*

$$\mathbb{E}\|\tilde{x}_k - x^*\|_2^2 < (1 - \theta)^k\|\tilde{x}_0 - x^*\|_2^2.$$

# 4 Barzilai-Borwein Step Size for SGD

In this section, we propose to incorporate the BB method to SGD (2). The BB method does not apply to SGD directly, because SGD never computes the full gradient $\nabla F(x)$. One may suggest to use $\nabla f_{i_{t+1}}(x_{t+1}) - \nabla f_{i_t}(x_t)$ to estimate $\nabla F(x_{t+1}) - \nabla F(x_t)$ when computing the BB step size using formula (6). However, this approach does not work well because of the variance of the stochastic gradients. The recent work by Sopyła and Drozda [18] suggested several variants of this idea to compute an estimated BB step size using the stochastic gradients. However, these ideas lack theoretical justifications and the numerical results in [18] show that these approaches are inferior to some existing methods.

The SGD-BB algorithm we propose in this paper works in the following manner. We call every $m$ iterations of SGD as one epoch. Following the idea of SVRG-BB, SGD-BB also uses the same step size computed by the BB formula in every epoch. Our SGD-BB algorithm is described as in Algorithm 2.

---

**Algorithm 2** SGD with BB step size (SGD-BB)

---

**Parameters**: update frequency $m$, initial step sizes $\eta_0$ and $\eta_1$ (only used in the first two epochs), weighting parameter $\beta \in (0, 1)$, initial point $\tilde{x}_0$
**for** $k = 0, 1, \cdots$ **do**
   **if** $k > 0$ **then**
      $\eta_k = \frac{1}{m} \cdot \|\tilde{x}_k - \tilde{x}_{k-1}\|_2^2 / |(\tilde{x}_k - \tilde{x}_{k-1})^\top (\hat{g}_k - \hat{g}_{k-1})|$
   **end if**
   $x_0 = \tilde{x}_k$
   $\hat{g}_{k+1} = 0$
   **for** $t = 0, \cdots, m-1$ **do**
      Randomly pick $i_t \in \{1, \ldots, n\}$
      $x_{t+1} = x_t - \eta_k \nabla f_{i_t}(x_t)$                                     $(*)$
      $\hat{g}_{k+1} = \beta \nabla f_{i_t}(x_t) + (1 - \beta)\hat{g}_{k+1}$
   **end for**
   $\tilde{x}_{k+1} = x_m$
**end for**

---

**Remark 3.** *We have a few remarks about SGD-BB (Algorithm 2).*

*(i) SGD-BB takes a convex combination of the $m$ stochastic gradients in one epoch as an estimation of the full gradient with parameter $\beta$. The performance of SGD-BB on different problems is not sensitive to the choice of $\beta$. For example, setting $\beta = 10/m$ worked well for all test problems in our experiments.*

*(ii) Note that for computing $\eta_k$ in Algorithm 2, we actually take the absolute value for the BB formula (6). This is because that unlike SVRG-BB, $\hat{g}_k$ in Algorithm 2 is not an exact full gradient. As a result, the step size generated by (6) can be negative. This can be seen from the following argument. Consider a simple case in which $\beta = 1/m$, approximately we have*

$$\hat{g}_k = \frac{1}{m} \sum_{t=0}^{m-1} \nabla f_{i_t}(x_t). \tag{11}$$

*It is easy to see that $\tilde{x}_k - \tilde{x}_{k-1} = -m\eta_{k-1}\hat{g}_k$. By substituting this equality into the equation for computing $\eta_k$ in Algorithm 2, we have*

$$\eta_k = (1/m) \cdot \|\tilde{x}_k - \tilde{x}_{k-1}\|^2 / |(\tilde{x}_k - \tilde{x}_{k-1})^\top (\hat{g}_k - \hat{g}_{k-1})|$$
$$= \frac{\eta_{k-1}}{\left| 1 - \hat{g}_k^\top \hat{g}_{k-1} / \|\hat{g}_k\|_2^2 \right|}. \tag{12}$$

*Without taking the absolute value, the denominator of (12) is $\hat{g}_k^\top \hat{g}_{k-1}/\|\hat{g}_k\|_2^2 - 1$, which is usually negative in stochastic settings.*

*(iii) Moreover, from (12) we have the following observations. If $\hat{g}_k^\top \hat{g}_{k-1} < 0$, then $\eta_k$ is smaller than $\eta_{k-1}$. This is reasonable because $\hat{g}_k^\top \hat{g}_{k-1} < 0$ indicates that the step size is too large and we need to shrink it. If $\hat{g}_k^\top \hat{g}_{k-1} > 0$, then it indicates that we should be more aggressive to take larger step size. Hence, the way we compute $\eta_k$ in Algorithm 2 is in a sense to dynamically adjust the step size, by*

*evaluating whether we are moving the iterates along the right direction. This kind of idea can be traced back to [11].*

Note that SGD-BB requires the averaged gradients in two epochs to compute the BB step size. Therefore, we need to specify the step sizes $\eta_0$ and $\eta_1$ for the first two epochs. From our numerical experiments, we found that the performance of SGD-BB is not sensitive to choices of $\eta_0$ and $\eta_1$.

### 4.1 Smoothing Technique for the Step Sizes

Due to the randomness of the stochastic gradients, the step size computed in SGD-BB may vibrate drastically sometimes and this may cause instability of the algorithm. Inspired by [13], we propose the following smoothing technique to stabilize the step size.

It is known that in order to guarantee the convergence of SGD, the step sizes are required to be diminishing. Similar as in [13], we assume that the step sizes are in the form of $C/\phi(k)$, where $C > 0$ is an unknown constant that needs to be estimated, $\phi(k)$ is a pre-specified function that controls the decreasing rate of the step size, and a typical choice of function $\phi$ is $\phi(k) = k + 1$. In the $k$-th epoch of Algorithm 2, we have all the previous step sizes $\eta_2, \eta_3, \ldots, \eta_k$ generated by the BB method, while the step sizes generated by the function $C/\phi(k)$ are given by $C/\phi(2), C/\phi(3), \ldots, C/\phi(k)$. In order to ensure that these two sets of step sizes are close to each other, we solve the following optimization problem to determine the unknown parameter $C$:

$$\hat{C}_k := \underset{C}{\operatorname{argmin}} \sum_{j=2}^{k} \left[ \log \frac{C}{\phi(j)} - \log \eta_j \right]^2. \tag{13}$$

Here we take the logarithms of the step sizes to ensure that the estimation is not dominated by those $\eta_j$'s with large magnitudes. It is easy to verify that the solution to (13) is given by $\hat{C}_k = \prod_{j=2}^{k} [\eta_j \phi(j)]^{1/(k-1)}$. Therefore, the smoothed step size for the $k$-th epoch of Algorithm 2 is:

$$\tilde{\eta}_k = \hat{C}_k/\phi(k) = \prod_{j=2}^{k} [\eta_j \phi(j)]^{1/(k-1)} / \phi(k). \tag{14}$$

That is, we replace the $\eta_k$ in equation $(*)$ of Algorithm 2 by $\tilde{\eta}_k$ in (14). In practice, we do not need to store all the $\eta_j$'s and $\hat{C}_k$ can be computed recursively by $\hat{C}_k = \hat{C}_{k-1}^{(k-2)/(k-1)} \cdot [\eta_k \phi(k)]^{1/(k-1)}$.

### 4.2 Incorporating BB Step Size to SGD Variants

The BB step size and the smoothing technique we used in SGD-BB (Algorithm 2) can also be used in other variants of SGD, because these methods only require the gradient estimations, which are accessible in all SGD variants. For example, when replacing the stochastic gradient in Algorithm 2 by the averaged gradients in SAG method, we obtain SAG with BB step size (denoted as SAG-BB). Because SAG does not need diminishing step sizes to ensure convergence, in the smoothing technique we just choose $\phi(k) \equiv 1$. The details of SAG-BB are given in the supplementary material.

## 5 Numerical Experiments

In this section, we conduct numerical experiments to demonstrate the efficacy of our SVRG-BB (Algorithm 1) and SGD-BB (Algorithm 2) algorithms. In particular, we apply SVRG-BB and SGD-BB to solve two standard testing problems in machine learning: logistic regression with $\ell_2$-norm regularization (LR), and the squared hinge loss SVM with $\ell_2$-norm regularization (SVM):

$$(\textit{LR}) \qquad \min_{x} F(x) = \frac{1}{n} \sum_{i=1}^{n} \log \left[ 1 + \exp(-b_i a_i^\top x) \right] + \frac{\lambda}{2} \|x\|_2^2, \tag{15}$$

$$(\textit{SVM}) \qquad \min_{x} F(x) = \frac{1}{n} \sum_{i=1}^{n} \left( [1 - b_i a_i^\top x]_+ \right)^2 + \frac{\lambda}{2} \|x\|_2^2, \tag{16}$$

where $a_i \in \mathbb{R}^d$ and $b_i \in \{\pm 1\}$ are the feature vector and class label of the $i$-th sample, respectively, and $\lambda > 0$ is a weighting parameter.

We tested SVRG-BB and SGD-BB on three standard real data sets, which were downloaded from the LIBSVM website[1]. Detailed information of the data sets are given in Table 1.

Table 1: Data and model information of the experiments

| Dataset | $n$ | $d$ | model | $\lambda$ |
|---|---|---|---|---|
| rcv1.binary | 20,242 | 47,236 | LR | $10^{-5}$ |
| w8a | 49,749 | 300 | LR | $10^{-4}$ |
| ijcnn1 | 49,990 | 22 | SVM | $10^{-4}$ |

## 5.1 Numerical Results of SVRG-BB

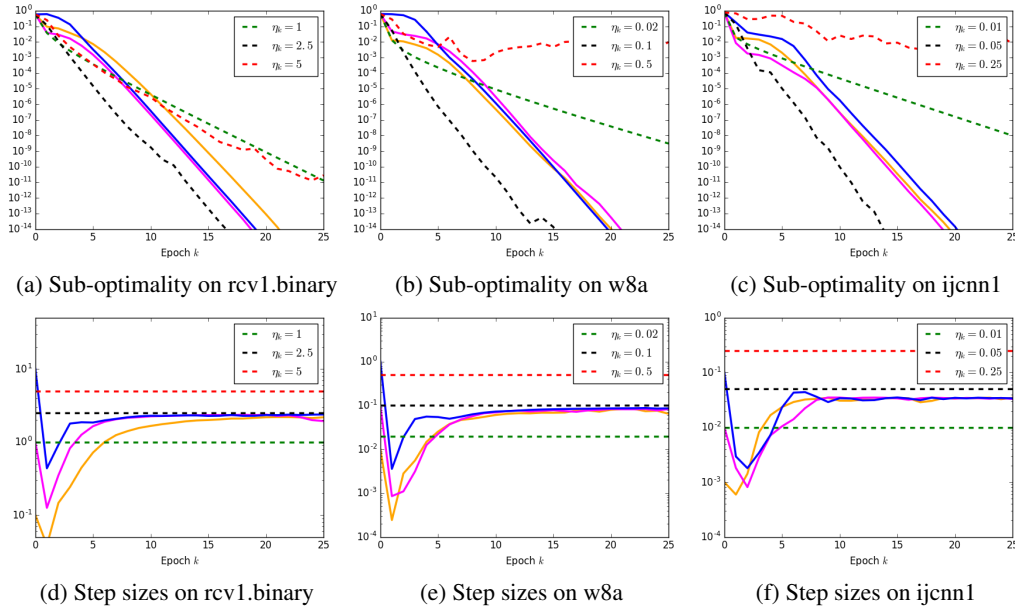

(a) Sub-optimality on rcv1.binary    (b) Sub-optimality on w8a    (c) Sub-optimality on ijcnn1

(d) Step sizes on rcv1.binary    (e) Step sizes on w8a    (f) Step sizes on ijcnn1

Figure 1: Comparison of SVRG-BB and SVRG with fixed step sizes on different problems. The dashed lines stand for SVRG with different fixed step sizes $\eta_k$ given in the legend. The solid lines stand for SVRG-BB with different $\eta_0$; for example, the solid lines in sub-figures (a) and (d) correspond to SVRG-BB with $\eta_0 = 10, 1, 0.1$, respectively.

In this section, we compare SVRG-BB (Algorithm 1) and SVRG with fixed step size for solving (15) and (16). We used the best-tuned step size for SVRG, and three different initial step sizes $\eta_0$ for SVRG-BB. For both SVRG-BB and SVRG, we set $m = 2n$ as suggested in [10].

The comparison results of SVRG-BB and SVRG are shown in Figure 1. In all sub-figures, the $x$-axis denotes the number of epochs $k$, i.e., the number of outer loops in Algorithm 1. In Figures 1(a), 1(b) and 1(c), the $y$-axis denotes the sub-optimality $F(\tilde{x}_k) - F(x^*)$, and in Figures 1(d), 1(e) and 1(f), the $y$-axis denotes the corresponding step sizes $\eta_k$. $x^*$ is obtained by running SVRG with the best-tuned step size until it converges. In all sub-figures, the dashed lines correspond to SVRG with fixed step sizes given in the legends of the figures. Moreover, the dashed lines in black color always represent SVRG with best-tuned step size, and the green and red lines use a relatively larger and smaller fixed step sizes respectively. The solid lines correspond to SVRG-BB with different initial step sizes $\eta_0$.

It can be seen from Figures 1(a), 1(b) and 1(c) that, SVRG-BB can always achieve the same level of sub-optimality as SVRG with the best-tuned step size. Although SVRG-BB needs slightly more epochs compared with SVRG with the best-tuned step size, it clearly outperforms SVRG with the

[1]www.csie.ntu.edu.tw/~cjlin/libsvmtools/.

other two choices of step sizes. Moreover, from Figures 1(d), 1(e) and 1(f) we see that the step sizes computed by SVRG-BB converge to the best-tuned step sizes after about 10 to 15 epochs. From Figure 1 we also see that SVRG-BB is not sensitive to the choice of $\eta_0$. Therefore, SVRG-BB has very promising potential in practice because it generates the best step sizes automatically while running the algorithm.

## 5.2 Numerical Results of SGD-BB

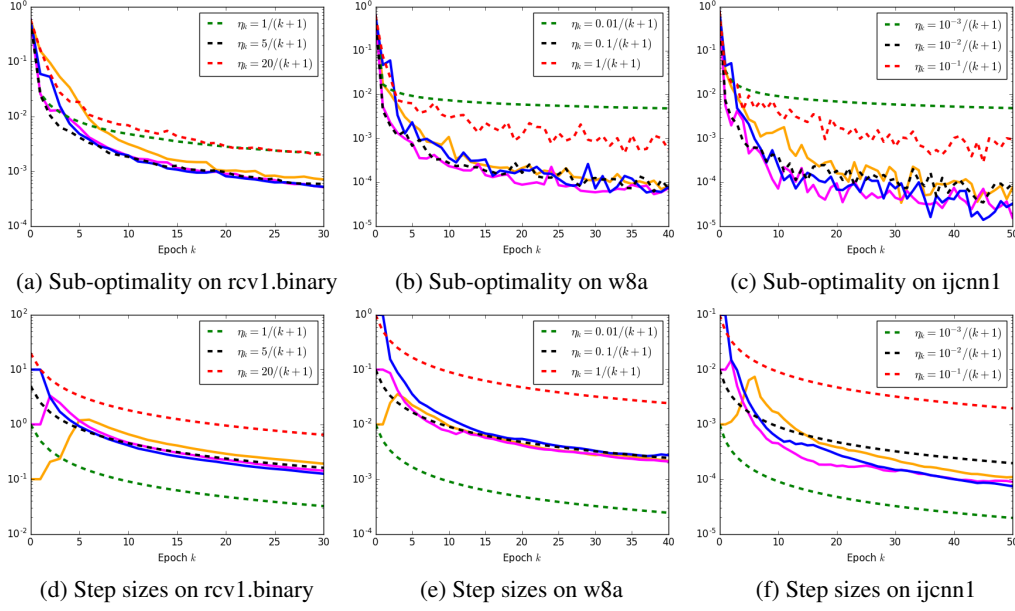

$$\text{(a) Sub-optimality on rcv1.binary} \qquad \text{(b) Sub-optimality on w8a} \qquad \text{(c) Sub-optimality on ijcnn1}$$

$$\text{(d) Step sizes on rcv1.binary} \qquad \text{(e) Step sizes on w8a} \qquad \text{(f) Step sizes on ijcnn1}$$

Figure 2: Comparison of SGD-BB and SGD. The dashed lines correspond to SGD with diminishing step sizes in the form $\eta/(k+1)$ with different constants $\eta$. The solid lines stand for SGD-BB with different initial step sizes $\eta_0$.

In this section, we compare SGD-BB with smoothing technique (Algorithm 2) with SGD for solving (15) and (16). We set $m = n$, $\beta = 10/m$ and $\eta_1 = \eta_0$ in our experiments. We used $\phi(k) = k + 1$ when applying the smoothing technique. Since SGD requires diminishing step size to converge, we tested SGD with diminishing step size in the form $\eta/(k+1)$ with different constants $\eta$. The comparison results are shown in Figure 2. Similar as Figure 1, the dashed line with black color represents SGD with the best-tuned $\eta$, and the green and red dashed lines correspond to the other two choices of $\eta$; the solid lines represent SGD-BB with different $\eta_0$.

From Figures 2(a), 2(b) and 2(c) we can see that SGD-BB gives comparable or even better sub-optimality than SGD with best-tuned diminishing step size, and SGD-BB is significantly better than SGD with the other two choices of step size. From Figures 2(d), 2(e) and 2(f) we see that after only a few epochs, the step sizes generated by SGD-BB approximately coincide with the best-tuned ones. It can also be seen that after only a few epochs, the step sizes are stabilized by the smoothing technique and they approximately follow the same decreasing trend as the best-tuned diminishing step sizes.

## 5.3 Comparison with Other Methods

We also compared our algorithms with many existing related methods. Experimental results also demonstrated the superiority of our methods. The results are given in the supplementary materials.

## Acknowledgements

Research of Shiqian Ma was supported in part by the Hong Kong Research Grants Council General Research Fund (Grant 14205314). Research of Yu-Hong Dai was supported by the Chinese NSF (Nos. 11631013 and 11331012) and the National 973 Program of China (No. 2015CB856000).

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
