[Supplementary Material · supplementary.pdf]

# Supplementary Materials: Barzilai-Borwein Step Size for Stochastic Gradient Descent

**Conghui Tan**
The Chinese University of Hong Kong
chtan@se.cuhk.edu.hk

**Shiqian Ma**
The Chinese University of Hong Kong
sqma@se.cuhk.edu.hk

**Yu-Hong Dai**
Chinese Academy of Sciences, Beijing, China
dyh@lsec.cc.ac.cn

**Yuqiu Qian**
The University of Hong Kong
qyq79@connect.hku.hk

## 1 Proofs

### 1.1 Proof of Lemma 1

Before proving Lemma 1, we first state the following lemma which is from [4] and will be useful in our proof.

**Lemma 2** (co-coercivity). *If $f(x) : \mathbb{R}^d \to \mathbb{R}$ is convex and its gradient is L-Lipschitz continuous, then*

$$\|\nabla f(x) - \nabla f(y)\|_2^2 \leq L(x-y)^\top (\nabla f(x) - \nabla f(y)), \quad \forall x, y \in \mathbb{R}^d.$$

Now we prove Lemma 1.

*Proof of Lemma 1.* Let $v_{i_t}^t = \nabla f_{i_t}(x_t) - \nabla f_{i_t}(\tilde{x}_k) + \nabla F(\tilde{x}_k)$ for the $k$-th epoch of SVRG-I or SVRG-BB. Then,

$$
\begin{aligned}
\mathbb{E}\|v_{i_t}^t\|_2^2 =& \mathbb{E}\|(\nabla f_{i_t}(x_t) - \nabla f_{i_t}(x^*)) - (\nabla f_{i_t}(\tilde{x}_k) - \nabla f_{i_t}(x^*)) + \nabla F(\tilde{x}_k)\|_2^2 \\
\leq& 2\mathbb{E}\|\nabla f_{i_t}(x_t) - \nabla f_{i_t}(x^*)\|_2^2 + 4\mathbb{E}\|\nabla f_{i_t}(\tilde{x}_k) - \nabla f_{i_t}(x^*)\|_2^2 + 4\|\nabla F(\tilde{x}_k)\|_2^2 \\
\leq& 2L\mathbb{E}\left[(x_t - x^*)^\top (\nabla f_i(x_t) - \nabla f_i(x^*))\right] + 4L^2\|\tilde{x}_k - x^*\|_2^2 + 4L^2\|\tilde{x}_k - x^*\|_2^2 \\
=& 2L(x_t - x^*)^\top \nabla F(x_t) + 8L^2\|\tilde{x}_k - x^*\|_2^2,
\end{aligned}
$$

where in the first inequality we used the inequality $(a-b)^2 \leq 2a^2 + 2b^2$ twice, in the second inequality we applied Lemma 2 to $f_{i_t}(x)$ and used the Lipschitz continuity of $\nabla f_{i_t}$ and $\nabla F$, and in the last equality we used the facts that $\mathbb{E}[\nabla f_{i_t}(x)] = \nabla F(x)$ and $\nabla F(x^*) = 0$.

In the next, we bound the distance of $x_{t+1}$ to $x^*$ conditioned on $x_t$ and $\tilde{x}_k$.

$$
\begin{aligned}
& \mathbb{E}\|x_{t+1} - x^*\|_2^2 \\
=& \mathbb{E}\|x_t - \eta_k v_{i_t}^t - x^*\|_2^2 \\
=& \|x_t - x^*\|_2^2 - 2\eta_k \mathbb{E}[(x_t - x^*)^\top v_{i_t}^t] + \eta_k^2 \mathbb{E}\|v_{i_t}^t\|_2^2 \\
=& \|x_t - x^*\|_2^2 - 2\eta_k (x_t - x^*)^\top \nabla F(x_t) + \eta_k^2 \mathbb{E}\|v_{i_t}^t\|_2^2 \\
\leq& \|x_t - x^*\|_2^2 - 2\eta_k (x_t - x^*)^\top \nabla F(x_t) + 2\eta_k^2 L(x_t - x^*)^\top \nabla F(x_t) + 8\eta_k^2 L^2\|\tilde{x}_k - x^*\|_2^2 \\
=& \|x_t - x^*\|_2^2 - 2\eta_k (1 - \eta_k L)(x_t - x^*)^\top \nabla F(x_t) + 8\eta_k^2 L^2\|\tilde{x}_k - x^*\|_2^2 \\
\leq& \|x_t - x^*\|_2^2 - 2\eta_k \mu (1 - \eta_k L)\|x_t - x^*\|^2 + 8\eta_k^2 L^2\|\tilde{x}_k - x^*\|_2^2 \\
=& [1 - 2\eta_k \mu (1 - \eta_k L)]\|x_t - x^*\|_2^2 + 8\eta_k^2 L^2\|\tilde{x}_k - x^*\|_2^2,
\end{aligned}
$$

where in the third equality we used the fact that $\mathbb{E}[v_{i_t}^t] = \nabla F(x_t)$, and in the second inequality we used the strong convexity of $F(x)$.

By recursively applying the above inequality over $t$, and noting that $\tilde{x}_k = x_0$ and $\tilde{x}_{k+1} = x_m$, we can obtain

$$\mathbb{E}\|\tilde{x}_{k+1} - x^*\|_2^2$$

$$\leq [1 - 2\eta_k\mu(1-\eta L)]^m \|\tilde{x}_k - x^*\|_2^2 + 8\eta_k^2 L^2 \sum_{j=0}^{m-1} [1 - 2\eta_k\mu(1-\eta L)]^j \|\tilde{x}_k - x^*\|_2^2$$

$$< \left[ (1 - 2\eta_k\mu(1-\eta L))^m + \frac{4\eta_k L^2}{\mu(1-\eta_k L)} \right] \|\tilde{x}_k - x^*\|_2^2$$

$$= \alpha_k \|\tilde{x}_k - x^*\|_2^2.$$

$\square$

## 1.2 Proof of Theorem 1

*Proof of Theorem 1.* Using the strong convexity of function $F(x)$, it is easy to obtain the following upper bound for the BB step size computed in Algorithm 1.

$$\eta_k = \frac{1}{m} \cdot \frac{\|\tilde{x}_k - \tilde{x}_{k-1}\|_2^2}{(\tilde{x}_k - \tilde{x}_{k-1})^\top (g_k - g_{k-1})}$$

$$\leq \frac{1}{m} \cdot \frac{\|\tilde{x}_k - \tilde{x}_{k-1}\|_2^2}{\mu\|\tilde{x}_k - \tilde{x}_{k-1}\|_2^2} = \frac{1}{m\mu}.$$

Similarly, by the $L$-Lipschitz continuity of $\nabla F(x)$, it is easy to obtain that $\eta_k$ is uniformly lower bounded by $1/(mL)$. Therefore, $\alpha_k$ in (8) can be bounded as:

$$\alpha_k \leq \left[ 1 - \frac{2\mu}{mL}\left(1 - \frac{L}{m\mu}\right) \right]^m + \frac{4L^2}{m\mu^2[1 - L/(m\mu)]}$$

$$\leq \exp\left\{ -\frac{2\mu}{mL}\left(1 - \frac{L}{m\mu}\right) \cdot m \right\} + \frac{4L^2}{m\mu^2[1 - L/(m\mu)]}$$

$$= \exp\left\{ -\frac{2\mu}{L} + \frac{2}{m} \right\} + \frac{4L^2}{m\mu^2 - L\mu},$$

Substituting (10) into the above inequality yields

$$\alpha_k < \exp\left\{ -\frac{2\mu}{L} + \log(1 - 2\theta) + \frac{2\mu}{L} \right\} + \frac{4L^2}{4L^2/\theta + L\mu - L\mu} = (1 - 2\theta) + \theta = 1 - \theta.$$

The desired result follows by applying Lemma 1. $\square$

## 2 SAG-BB Algorithm

The stochastic averaged gradient method with BB step size (SAG-BB) is shown in Algorithm 3. Because we choose $\phi(k) \equiv 1$ in the smoothing technique, it is easy to see the smoothed step size $\tilde{\eta}_t$ is equal to the geometric mean of all previous BB step sizes.

## 3 Sensitivity of SVRG-BB to $m$

In this section, we empirically study the sensitivity of SVRG-BB to its inner iteration number: $m$. We tested both SVRG-BB and SVRG for several choices of $m$. The initial step sizes $\eta_0$ of SVRG-BB were randomly chosen, and we used the best-tuned fixed step sizes for SVRG. The comparison results are shown in Figure 3.

From Figure 3, we can see that for the same $m$, SVRG-BB usually needs slightly more number of passes over the data, to obtain the same sub-optimality compared to SVRG with best-tuned step size.

---

**Algorithm 3** SAG with BB step size (SAG-BB)

---

**Parameters**: update frequency $m$, initial step sizes $\eta_0$ and $\eta_1$ (only used in the first two epochs), weighting parameter $\beta \in (0, 1)$, initial point $\tilde{x}_0$

$y_i = 0$ for $i = 1, \ldots, n$

**for** $k = 0, 1, \cdots$ **do**

   **if** $k > 0$ **then**

      $\eta_k = \frac{1}{m} \cdot \|\tilde{x}_k - \tilde{x}_{k-1}\|_2^2 / |(\tilde{x}_k - \tilde{x}_{k-1})^\top (\hat{g}_k - \hat{g}_{k-1})|$

      $\tilde{\eta}_k = \left( \prod_{j=2}^{k} \eta_j \right)^{\frac{1}{k-1}}$                 ▷ smoothing technique

   **end if**

   $x_0 = \tilde{x}_k$

   $\hat{g}_{k+1} = 0$

   **for** $t = 0, \cdots, m - 1$ **do**

      Randomly pick $i_t \in \{1, \ldots, n\}$

      $y_{i_t} = \nabla f_{i_t}(x_t)$

      $x_{t+1} = x_t - \frac{\eta_k}{n} \sum_{i=1}^{n} y_i$                 ▷ SAG update

      $\hat{g}_{k+1} = \beta \nabla f_{i_t}(x_t) + (1 - \beta)\hat{g}_{k+1}$

   **end for**

   $\tilde{x}_{k+1} = x_m$

**end for**

---

| (a) rcv1.binary | (b) w8a | (c) ijcnn1 |

Figure 3: Comparison of SVRG-BB and best-tuned SVRG over different values of $m$. The $x$-axes all denote the number of passes over the data, and the $y$-axes all denote the sub-optimality $F(x_k) - F(x^*)$. In each sub-figure, the solid lines stand for SVRG-BB, while dashed lines with the same colors stand for SVRG with the same $m$.

However, after a few number of passes over the data, SVRG-BB reaches the same sub-optimality as SVRG with best-tuned step size. These results are consistent with our previous experimental results in Section 5.1 of the paper. Further more, when $m$ is small, the performance of SVRG-BB is usually close to or even superior than SVRG with best-tuned step size. This is natural because step sizes are updates more frequently when $m$ is small, and the initialization time spent by SVRG-BB in adjusting step sizes is thus almost ignorable.

## 4 Numerical Comparisons with Other Methods

In this section, we compare our SVRG-BB (Algorithm 1,) SGD-BB (Algorithm 2) and SAG-BB (Algorithm 3) with several existing methods: MISO-2 [3], AdaGrad [1], Adam [2], vSGD [6], SAG with line search (denoted as SAG-L) [5], and a stochastic quasi-Newton method: oLBFGS [7]. For SVRG-BB, we set $m = 2n$; while for both SGD-BB and SAG-BB, we set $m = n$ and $\beta = 10/m$. Because these methods have very different per-iteration complexity, we compare their CPU times needed to achieve the same sub-optimality.

Figures 4(a), 4(b) and 4(c) show the comparison results of SVRG-BB and MISO-2. We see from these figures that SVRG-BB is faster than MISO-2.

Figures 4(d), 4(e) and 4(f) show the comparison results of SGD-BB and AdaGrad. From these figures we see that AdaGrad usually has a very quick start, but in many cases the convergence becomes slow

| rcv1.binary | w8a | ijcnn1 |
|---|---|---|
| (a) MISO-2 versus SVRG-BB | (b) MISO-2 versus SVRG-BB | (c) MISO-2 versus SVRG-BB |
| (d) AdaGrad versus SGD-BB | (e) AdaGrad versus SGD-BB | (f) AdaGrad versus SGD-BB |
| (g) Others versus SGD-BB | (h) Others versus SGD-BB | (i) Others versus SGD-BB |
| (j) SAG-L versus SAG-BB | (k) SAG-L versus SAG-BB | (l) SAG-L versus SAG-BB |

Figure 4: Comparison between SVRG-BB, SGD-BB and SAG-BB with some existing related methods. The $x$-axes all denote the CPU time (in seconds). The $y$-axes all denote the sub-optimality $F(x_k) - F(x^*)$. In the first row, solid lines stand for SVRG-BB, and dashed lines stand for MISO-2; In the second and third rows, solid lines stand for SGD-BB, and dashed lines stand for AdaGrad, Adam, vSGD and oLBFGS respectively; In the final row, solid lines denote SAG-BB, while dashed ones stand for SAG with line search.

in later iterations. Besides, AdaGrad is still somewhat sensitive to the initial step sizes. Especially, when a small initial step size is used, AdaGrad is not able to increase the step size to a suitable level. As a contrast, SGD-BB converges very fast in all three tested problems, and it is not sensitive to the initial step size $\eta_0$.

In Figures 4(g), 4(h) and 4(i), we further compared SGD-BB with Adam, vSGD and oLBFGS. Again, we observed that our SGD-BB is faster than these compared methods.

Figures 4(j), 4(k) and 4(l) show the comparison results of SAG-BB and SAG-L. From these figures we see that the SAG-L is quite robust and is not sensitive to the choice of $\eta_0$. However, SAG-BB is faster than SAG-L to reach the same sub-optimality on the tested problems.