[Reviews · NeurIPS 2016]

Reviewer 1

Summary

This paper considers the question of adapting the step-size in Stochastic Gradient descent (SGD) and some of its variants. It proposes to use the Barzilai Borwein (BB) method to automatically compute step-sizes in SGD and stochastic variance reduced gradient (SVRG) instead of relying on predefined fixed (decreasing) schemes. For SGD a smoothing technique is additionally used.

Qualitative Assessment

The paper addresses an important question for SGD type of algorithms. The BB method is first implemented within the SVGR. The simulation are convincing is that the optimal step-size is learned after an adaptation phase. I am however wondering why in Figure 1 there is this strong overshoot towards much too small step-sizes in the first iterations. It looks suboptimal. For the implementation within SGD, the author(s) need to introduce a smoothing technique. My concern is that within the smoothing formula they reintroduce a deterministic non-adaptive decrease. They explicitly reintroduce a decrease in 1/k+1. Hence the proposed adaptation scheme seems to present the same drawbacks as a predefined scheme. Other comments: In lemma 1, the expectation should be a conditional expectation.

Confidence in this Review

2-Confident (read it all; understood it all reasonably well)


Reviewer 2

Summary

The authors give a new analysis of SVRG. It allows using "Option I" (taking the final iterate of the inner iteration), as is done in practice. They also propose to use a scaled version of Barzilai-Borwein to set the step-sizse for SVRG (and heuristically argue that this could also be useful for classic stochastic gradient methods too). Their experiments show that this adaptive step-size is competitive with fixed step-sizes.

Qualitative Assessment

Note that I increased my score in light of the experiments discussed in the author response. I previously reviewed this paper for ICML. Below I've included some quotes from my ICML review that are still relevant. But first, I'll comment on some of the changes and lack of changes after the previous round of reviewing: 1. The authors have removed most of the misleading statements and included extra references and discussion, which I think makes the paper much better. 2. One reviewer brought up how the quadratic dependence on some of the problem constants is inferior to existing results. I'm ok with this as having an automatic step-size is a big advantage, but the paper should point out explicitly that the bound is looser. (This reviewer also pointed out that achieving a "bad" rate under Option I is easy to establish, although in this case I agree with the authors that this contribution is novel). 3. The paper is *still* missing an empirical comparison with the existing heuristics that people use to tune the step-size. The SAG line-search is now discussed in the introduction (and note that this often works better than the "best tuned step size" since it can increase the step-size as it approaches the solution) but there is no experimental comparison and the MISO line-search heuristic is still not even discussed. To me this is a strange omission: if the proposed method actually works better than these methods then including these comparisons only makes the paper stronger. Even if the proposed method works similarly to existing methods, it still makes the paper stronger because it shows that we now have a theoretically-justified way to get the same performance. Not including these comparisons is not only incomplete scholarship, but it makes the reader think there is something to hide. (I'm not saying there is something to hide, I'm just saying there are only good reasons to include these experiments and only bad reasons not to!) 4. One reviewer pointed out a severe restriction on the condition number in the previous submission, which has been fixed. --- Comments from old review --- Summary: The authors give a new analysis of SVRG. It allows using "Option I" (taking the final iterate of the inner iteration), as is done in practice. They also propose to use a scaled version of Barzilai-Borwein to set the step-sizse for SVRG (and heuristically argue that this could also be useful for classic stochastic gradient methods too). Their experiments show that this adaptive step-size is competitive with fixed step-sizes. Clarity: The paper is very clearly-written and easy to understand (though many grammar issues remain). Significance: Although several heuristic adaptive step-size strategies exist in the literature, this is the first theoretically-justified method. It sill depends on constants that we don't know in general, but I believe is a step towards black-box SG methods. Details: Independent of the SVRG/SG results, the authors give a nice way to bound the step-size for the BB method. Normally, BB leads to a much faster rate than using a constant step-size, but in the SVRG setting your theory/experiments are just showing that it does as well as the best step-size (which is good, but it isn't better than the best step size). Finally, the paper would be much stronger if it compared to the two existing strategies that are used in practice: 1. The line-search of Le Roux et al. where they increase/decrease an estimate of L. 2. The line-search of Mairal where he empirically tries to find the best step-size. However, I don't think that the proposed approach would actually work better than both of these methods (but these older approaches don't have any theory).

Confidence in this Review

3-Expert (read the paper in detail, know the area, quite certain of my opinion)


Reviewer 3

Summary

In this paper the authors incorporate the Barzilai-Borwein (BB) step-size for variants of stochastic gradient descent such as SGD and SVRG. It is also introduced to SAG in the appendix. It is proven that the SVRG with BB step size converges linearly in expectation on mu-strongly convex function with Lipschitz continuous gradient as long as m is chosen sufficiently large, where m is an integer such that the full gradient is computed every m iterations. Experimental results show the proposed SVRG with BB step size tends to converge the step size towards the optimally tuned step size, resulting in a slightly low but similar convergence rate on three data sets. The results of the other variants of SGD with BB step size looks also promising.

Qualitative Assessment

In the SVRG-I, which is claimed in this paper to be more efficient in practice than SVRG-II, two parameters, an integer m and the step size eta, need to be tuned to achieve linear convergence. The contribution of the paper is that it makes SVRG easier to use by introducing BB step size in the sense that we do not need to tune the step size anymore as long as m is taken sufficiently large. This paper will be improved if the sensitivity of m is studied empirically. The theorem tells that one can achieve the linear convergence with the rate of convergence of 1 - theta in terms of the number of epochs. If the convergence rate in terms of epoch is fixed, the time complexity will increase linearly in terms of m. If we set m twice, we expect twice slow convergence. If this is the case, tuning of m is as sensitive as tuning of the step size, and is not very nice.

Confidence in this Review

2-Confident (read it all; understood it all reasonably well)


Reviewer 4

Summary

The authors propose an automatic step size scheme using the Barzilai-Borwein method for SGD and SVRG. Linear convergence analysis is provided for SVRG-BB and SVRG-I (option 1 of SVRG where the current iterate is used as the mark iterate on the next epoch). A smoothing technique is required on top of the BB step size for SGD-BB to converge, but no convergence analysis is provided for SGD-BB. The authors compare SVRG-BB to SVRG with a constant step size, and SGD-BB to SGD with a decreasing step size scheme, and find that the BB step sizes have a comparable rate of convergence to the best tuned step size for both SVRG and SGD.

Qualitative Assessment

EDIT: I have increased my ratings based on the author's response. This paper addresses an important problem: that of automatically determining a step size sequence for stochastic methods like SGD. The authors use a well-known method, the Barzilai-Borwein method, to automatically determine step sizes for SGD and SVRG. While the results are promising, there are two major shortcomings of this paper: 1. No convergence rate analysis for SGD-BB is provided. While the authors rightly mention that a decreasing step size is required for convergence of the SGD method, it is important to theoretically study how the BB step size affects the convergence rate of SGD. Can the authors comment on this? Moreover, the optimization function used for smoothing the SGD step sizes (equation 13) seem quite heuristic, with no theoretical justification. 2. The numerical experiments performed are not very extensive. The authors only perform experiments on binary classification models. It would be interesting to see how the step sizes perform on regression problems, and non-convex problems like neural networks. Moreover, the authors do not compare SGD-BB to other automatic step size methods for SGD (see for references [1] and [2]). There is no discussion in the paper about how the BB step sizes compare to existing automatic step size methods. For example, is this method faster than the other proposed methods, or does it use lesser memory? The convergence analysis for SVRG-BB and SVRG-I is simple, but interesting. However, the application of a BB step size on SVRG is not a big contribution since SVRG calculates true gradients after each epoch. Thus, the BB step size can be trivially applied on SVRG, as demonstrated in the paper. Overall, while the paper shows good results and is addressing a very important problem in methods like SGD, I think it requires a bit more work. In particular, the paper would benefit from a convergence rate proof for SGD-BB, as well as slightly more extensive experiments. Some minor comments: 1. Typo in line 13 of supplementary: \eta -> \eta_k. 2. Bounds in line 9 of supplementary can be improved using the same technique as in the original SVRG paper 3. How sensitive is SGD-BB to m? 4. In SVRG-BB, the step sizes initially seem to be quite low. This results in slightly slower convergence at first. Do the authors have any intuition about why this might be happening? If the initial slowdown can be avoided, SVRG-BB might be able to match the performance of the best-tuned step size for SVRG. [1] Mahsereci, Maren, and Philipp Hennig. "Probabilistic line searches for stochastic optimization." Advances In Neural Information Processing Systems. 2015. [2] Schaul, Tom, Sixin Zhang, and Yann LeCun. "No more pesky learning rates." ICML (3) 28 (2013): 343-351.

Confidence in this Review

3-Expert (read the paper in detail, know the area, quite certain of my opinion)


Reviewer 5

Summary

This paper proposes SGD/SVRG using the Barzilai-Borwein (BB) method to tune step size automatically. The paper shows the linear convergence not only of SVRG-BB but also of SVRG with option-I whose proof has been missing in the literature. In experiments, SVRG-BB starting from several step sizes achieves competitive performance compared to best tuned SVRG.

Qualitative Assessment

The paper is well written and I can understand the motivation to use the BB method. The analyses are correct. Although the contribution is rather incremental, the paper is useful to community because it complements the proof of SVRG-I and provides highly practical methods. It would be nice if the author can give the theoretical guarantee to use BB method for vanilla SGD and other stochastic methods.

Confidence in this Review

2-Confident (read it all; understood it all reasonably well)


Reviewer 6

Summary

This work proposes to use Barzilai-Borwein method to compute learning rate for stochastic gradient method and its variants. It provides the linear convergence proof of SVRG-BB method, however, there is not any convergence analysis about SGD-BB method.

Qualitative Assessment

This work proposes a step size learning method for SGD and its variants. This kind of work is missing for SVRG, while there are many works about step size learning for SGD. Here is a list of my concerns: 1. In Fig 1 (d,e,f), it is interesting to see that the BB step size converges to a constant value. However, in the paper: Allen-Zhu, Zeyuan, and Elad Hazan. "Variance reduction for faster non-convex optimization.", Fig 5 shows that a best tuned decreasing learning rate has faster convergence rate for SVRG. Can you explain why it converges to a constant? 2. There are many step size learning works for SGD. In the experiment of SGD-BB section, I think there should be more comparing methods, like Adam, adagrad.

Confidence in this Review

2-Confident (read it all; understood it all reasonably well)